# A Narrative Review Exploring Attention Deficit/Hyperactivity Disorder in Patients with Early Psychosis

**DOI:** 10.3390/brainsci14030190

**Published:** 2024-02-20

**Authors:** Temi Toba-Oluboka, Kara Dempster

**Affiliations:** 1Nova Scotia Early Psychosis Program, Nova Scotia Health Authority, Halifax, NS B3H 2E2, Canada; temi.toba-oluboka@nshealth.ca; 2Department of Psychiatry, Dalhousie University, Halifax, NS B3H 4R2, Canada

**Keywords:** ADHD, early psychosis, schizophrenia, stimulant treatment, narrative review

## Abstract

While both Attention deficit hyperactivity disorder (ADHD) and schizophrenia are considered to have neurodevelopmental origins with associated impairments in executive functioning, there is a paucity of clinical guidelines pertaining specifically to this comorbidity. We sought to summarize the existing literature on ADHD in early psychosis patients, focusing on issues that would be most relevant to clinical practice. For this narrative review, we completed a search on PubMed and PsycINFO with 22 papers meeting criteria for review. Overall, it appears that a diagnosis of ADHD in childhood increases the risk of the subsequent development of a primary psychotic disorder. This risk may be modified by higher rates of substance use and could be related to shared premorbid risk factors for both conditions, such as obstetrical complications. Stimulant use has been associated with the onset of psychotic symptoms in some individuals, but it is unclear whether certain subgroups are more susceptible. Despite the fact that these two conditions co-occur relatively frequently, there is currently a lack of objective diagnostic tests for ADHD specific to populations with primary psychotic disorders, and a paucity of evidence on whether stimulants are effective for ADHD symptoms in this sub-group. Future research is warranted to investigate whether stimulant treatment confers any additional risks for symptom exacerbation in individuals with primary psychotic disorders taking antipsychotic maintenance treatment.

## 1. Introduction

Schizophrenia has been conceptualized as a neurodevelopmental disorder with an onset typically during adolescence and early adulthood [1]. Positive psychotic symptoms include hallucinations, delusions, disorganized speech and behaviour, and negative symptoms. Though not included in formal diagnostic criteria, cognitive impairment and executive functioning deficits are common in individuals with psychotic disorders [2,3]. Executive functioning impairments tend to be present prior to the development of overt psychotic symptoms and significantly impact activities of daily living. Furthermore, while antipsychotic medication treatment generally treats positive psychotic symptoms, medications blocking dopamine generally do not improve cognitive deficits, and if anything, can exacerbate them [4]. 

Attention deficit hyperactivity disorder (ADHD) is a relatively common neurodevelopmental disorder associated with symptoms of hyperactivity, inattention, and impulsivity. Individuals with ADHD experience executive functioning impairments such as difficulties or inabilities to sustain attention, organizing tasks, and inhibition control. Within early psychosis care, it is common for patients to experience distress related to impairments in executive functioning, many of whom believe their symptoms could be due to ADHD. This situation can present challenges clinically as it is difficult to objectively evaluate the extent of executive functioning impairments in an office setting. Furthermore, the use of stimulants, which are the first line treatment for ADHD, has been noted to increase the risk of positive psychotic symptoms, like hallucinations and delusions [5]. In practice, it is challenging to determine whether cognitive impairments are secondary to a primary psychotic disorder, comorbid ADHD, both, or some combination of other factors. Many clinicians are hesitant to prescribe stimulant treatment to individuals with psychosis given the association between stimulant use and the development of first episode psychosis (FEP) [6]. There is very little evidence investigating whether the use of a stimulant in an individual with early psychosis and ADHD might be expected to worsen positive symptoms of psychosis, or whether stimulants lead to improvements in ADHD symptoms in this population. The aim of this narrative review was to review and summarize existing literature on ADHD in early psychosis patients. 

## 2. Materials and Methods

Following the SANRA guidelines, we completed a literature search on PubMed and PsycINFO. Search terms were created with the assistance of a librarian: (ADHD OR “attention deficit disorder” OR “attention deficit hyperactivity disorder”) AND (psychosis OR psychotic OR schizophrenia) AND (early OR “first episode”). The search was conducted between 27 September and 30 October 2023 and we included studies that had been published up to October 2023. The last search was conducted on 30 October 2023. We used Covidence to screen and extract studies that met our inclusion criteria, as listed below. For papers to be included in this review, they had to meet the following inclusion criteria:
In research studies, participants must have a diagnosis of either psychosis, schizophrenia or ADHD;In comparison studies, studies must explore or identify similarities or differences between ADHD and psychosis;Papers must be written in English or French.

Two reviewers (TT-O and KD) independently screened titles and abstracts to exclude studies that did not meet the above inclusion criteria. Our initial search yielded 1391 papers, 289 duplicates were removed prior to title and abstract screening. Out of 1102 studies, 1045 studies were excluded during the title and abstract screening stage. A total of 57 studies then proceeded to full text screening. Out of these 57 studies, 35 were excluded (wrong outcome (*n* = 13), wrong patient population (*n* = 6), paediatric population (*n* = 5), wrong comparator (*n* = 4), wrong intervention (*n* = 2), wrong indication (*n* = 2), wrong setting (*n* = 1), written in a language other than English or French (*n* = 1), and a book chapter (*n* = 1)). Any disagreements between reviewers were discussed and resolved by reaching a consensus. A total of 22 studies were then included for review [7,8,9,10,11,12,13,14,15,16,17,18,19,20,21,22,23,24,25,26,27,28]. Please see Figure 1 of the schematic outlining our screening process. 

## 3. Results

### 3.1. Differences in Executive Functioning Profiles Help Differentiate ADHD from Cognitive Symptoms of Schizophrenia

While the cognitive profiles of both ADHD and early phase psychosis can overlap, there are some patterns of impaired cognition that can point towards one diagnosis over the other. While in depth neuropsychological testing is not readily accessible in many locations, some knowledge of the underlying patterns of cognitive impairment specific to each disorder provides important information that may help guide diagnosis. In our review nine papers investigated specific cognitive profiles of ADHD and early phase psychosis [12,13,14,15,18,20,21,22,28].

Sanchez-Gistau et al. (2023) investigated cognitive biases in ADHD and patients with FEP [12]. The study sample included 31 FEP patients who met criteria for childhood ADHD, 91 FEP patients without ADHD and 26 healthy controls (HC) [12]. The FEP patients with ADHD had significantly higher scores than FEP patients without ADHD and HC groups on the cognitive biases questionnaire for psychosis (CBQp), a self-report questionnaire measuring five cognitive biases (intentionalising, catastrophizing, dichotomous thinking, jumping to conclusions, and emotional reasoning) [12]. Additionally, this group had significantly higher scores on the CBQp themes of threatening events and anomalous perception [12]. FEP patients with ADHD showed greater intentionalising, dichotomous thinking catastrophism and emotional reasoning bias scores in comparison with the healthy control group [12]. Overall, this study found that cognitive biases were most prominent in FEP patients with ADHD relative to those without a childhood diagnosis of ADHD, with the healthy control group having the lowest scores [12].

Groom et al. (2008) investigated the sensitivity and specificity of cognitive trait markers for schizophrenia. The authors compared 30 adolescents with schizophrenia spectrum disorders, 36 non-psychotic siblings, 72 healthy controls, and 27 adolescents with ADHD [13]. This study found significant impairments in both the schizophrenia patients and their siblings on measures of intellectual ability, verbal memory, executive function, and sustained attention [13]. The schizophrenia patients were impaired on more measures in comparison with the siblings group and showed a larger degree of impairment in comparison with the heathy control group [13]. In the comparison with the ADHD and non-psychotic siblings groups, the ADHD group showed significant impairment in the shapes condition on the continuous performance test-identical pairs (CPT-IP) while the non-psychotic siblings group did not show impairment. Alternatively, non-psychotic siblings showed significantly increased response initiation speed on the Hayling sentence completion test (HSCT) while the ADHD group did not [13]. 

Øie et al. (2020) compared longitudinal cognitive outcomes in 10 early onset schizophrenia (EOS) patients, 19 ADHD patients and 26 healthy controls [14]. Participants in the EOS group had significant stagnation or deterioration on the cognitive composite score from baseline to time 2 (13 year follow-up) in comparison with the healthy control group [14]. The EOS group showed the most positive improvement from time 2 to time 3 (23–25 year follow up) [14]. At all three time points, the EOS group were most impaired on the cognitive composite score compared with the ADHD and healthy control groups [14]. In the ADHD group, the authors observed cognitive maturation with performance improving longitudinally in comparison with healthy controls [14]. 

In their 2011 study, Øie et al. investigated the relationship between baseline neurocognition and longitudinal changes in intellectual testing and functional outcomes in a 13 year follow-up study [20]. The authors compared 15 EOS patients, 19 ADHD patients, and 30 healthy controls [20]. In comparison with the healthy controls, both patient groups were found to have impaired functional outcomes [20]. For the EOS group, processing speed, selective attention, executive function and verbal memory measured at baseline were associated with reduced functional outcome longitudinally [20]. Lower scores on neurocognitive measures at baseline in both the ADHD and EOS groups were related to worse social functioning at the 13 year follow-up [20]. 

A study conducted by Jepsen et al. (2018) compared 29 individuals with early onset first episode schizophrenia, 29 individuals with ADHD, and 45 healthy controls [22]. The authors found that both the ADHD group and the EOS group shared significantly increased motor, attentional and non-planning straits of impulsivity [22]. The authors found that ADHD participants showed a significantly increased reflection impulsivity (an impaired ability to gather and evaluate information prior to decision making) [22].

Another study by Øie et al. (1999) compared the memory functions of 19 patients with schizophrenia, 20 ADHD patients and 30 healthy controls. Compared with healthy controls, the schizophrenia group showed impairments in most measures of memory [15]. Additionally, schizophrenia patients showed deficits in long term verbal recall but not in long term verbal recognition [15]. In comparison with the ADHD group, the schizophrenia patients performed worse than the ADHD group on visual memory tasks [15]. Compared with the patients with schizophrenia, the ADHD group showed more impairment on working memory tasks that focused on distractibility [15]. 

Egeland investigated the frequency of various measures of attention in first episode schizophrenia (FES) [18]. The study compared three groups, 27 FES patients, 55 individuals with comorbid ADHD and schizophrenia, and 20 patients with ADHD-inattentive subtype [18]. The study found that 47% of the FES patients were classified as having a severe impairment in shifting attention [18]. Additionally, 7% of FES patients had severe impairment in sustained attention [18]. In the ADHD-inattentive group, 78% of participants were severely impaired in shifting attention and 25% had a severe increase in impulsivity [18]. For the comorbid group, 50% of participants were impaired in shifting attention and impulsivity [18]. When comparing FES and the ADHD-inattentive group, more ADHD patients were severely impaired in shifting and sustained attention than patients in the FES group [18]. Fewer patients with FES without ADHD were impaired in focused and shifting attention in comparison with patients in the ADHD-inattentive group [18]. For sustained attention, the frequency of impairment was higher in the ADHD-inattentive group than the comorbid group and FES group [18]. The comorbid group did not differ from the FES patients on any measures of attention [18]. A total of 21% of the FES performed normally on all measures of attention [18]. The authors concluded that attention tests are not sensitive enough to characterize attention deficits in daily functioning [18].

Øie et al. investigated neurocognitive status in a 13-year follow-up study of 15 schizophrenia patients, 19 ADHD patients, and 30 healthy controls [21]. The authors found a significant decline in both verbal memory and learning in the schizophrenia patients at follow-up [21]. This group also showed a lack of improvement over time in attention and processing speed after 13 years [21]. The ADHD group showed significant improvement in attention at follow-up in comparison with healthy controls and schizophrenia patients [21].

A study by Sanchez-Gistau et al. (2020) compared the clinical characteristics and cognitive profiles of FEP patients with and without a history of childhood ADHD [28]. The study included 33 FEP patients with childhood ADHD, 100 FEP patients without ADHD and 65 healthy controls [28]. Both FEP groups performed significantly worse on domains testing processing speed, attention vigilance, executive function, verbal memory and social cognition. Both FEP groups did not differ in psychopathology, however, the FEP group with childhood ADHD were more likely to use tobacco and cannabis [28]. Looking at the severity of cognitive performance, FEP patients with a history of ADHD in childhood had greater impairments relative to FEP patients without ADHD [28]. 

### 3.2. Neurobiological Differences between ADHD and Schizophrenia

Investigating underlying neurobiological correlates neurodevelopmental disorders may provide useful information on shared risk factors and in the future and could be used for diagnosis and treatment selection. While there have been extensive neuroimaging studies investigating each disorder separately, only one study investigating both came up in our review. This study used transcranial magnetic stimulation (TMS) to examine cortical excitability in 25 FES patients, 28 ADHD patients and 41 healthy controls [7]. This study found that both the FES and ADHD groups demonstrated a prolonged contralateral silent period (CSP) in comparison with the healthy control group [7]. Additionally, the two patient groups had cortical disinhibition in comparison with the healthy controls. An enhanced intracortical facilitation (ICF) was found in the ADHD group in the left hemisphere in comparison with healthy controls [7]. In the FES group, an enhanced ICF at 15 ms over the right hemisphere and a reduced ICF at 7 ms over the left hemisphere were found to be significantly different from the ADHD group [7]. Both patient groups had a significantly different pattern of the hemispheric distribution of the inhibitory CSP and faciliatory ICF in comparison with the healthy control group [7]. Overall, while the ADHD and FES groups had a similar pattern of cortical disinhibition, there was a distinct hemispheric distribution of some facilitatory parameters between the ADHD and FES patients [7].

### 3.3. Comorbid ADHD and Schizophrenia

ADHD and early phase psychosis can co-occur. Information on the frequency of this co-occurrence is useful and could inform clinical practice. In our review, three papers discuss the prevalence of comorbid ADHD and early phase psychosis [9,17,28].

Marwaha et al. (2015) used data from the Adult Psychiatric Morbidity Survey, a household survey conducted in England to examine the relationship between ADHD symptoms and psychosis [9]. The survey had a total of 7403 respondents [9]. The authors found that higher levels of ADHD symptoms in adults were associated with paranoid ideation and auditory hallucinations [9]. Using scores from the ADHD self-report scale (ASRS), an association between probable ADHD and probable psychosis was found [9]. The post-hoc analysis found that an ASRS score of five or above indicated a significant association between ADHD caseness and paranoid ideation, auditory hallucinations however, this association was not found with probable psychosis. In their mediation analysis, illicit use of amphetamines was found to be associated with paranoid ideation [9]. 

In a nation-wide cohort study, Strålin et al. (2018) investigated comorbid disorders in FEP [17]. In this sample of 2091 FEP patients, 170 (8.1%) cases of comorbid ADHD were identified based on the ICD-10 diagnoses made at the time of psychiatric hospitalization [17]. The comorbid ADHD group had a significantly higher proportion of males in comparison with the rest of the cohort [17]. A comorbid diagnosis of substance use disorders (SUD) were more common in FEP patients with ADHD [17]. Additionally, self-harm was significantly more common in the patients with comorbid ADHD a year before the first hospitalization and during their first hospitalization [17]. Psychostimulants in this subpopulation were less common two years post-first hospitalization (41%) in comparison with the year prior to the first hospitalization (66%) [17]. Hospitalizations for SUD after discharge (23%) were more common in those with ADHD in comparison with the rest of the cohort (11%). Lastly, hospitalizations for self-harm were significantly more common in the ADHD group [17].

A study by Sanchez-Gistau et al. (2020) also investigated the rates of childhood ADHD in 133 FEP patients and 65 health controls [28]. To assess for ADHD, participants underwent an assessment using the Diagnostic interview for ADHD in adults (DIVA) with a psychiatrist [28]. The study found that 24.8% of their study sample met criteria for childhood ADHD. In their sample of FEP patients that met criteria for childhood ADHD, 36.4% had previously been diagnosed with ADHD in their childhood or adolescence [28]. However, only 41.7% of these patients received medication treatment for their ADHD after their onset of psychotic symptoms [28].

### 3.4. Diagnosis of ADHD in Childhood and the Risk of Developing Psychosis Later On

A diagnosis of ADHD in childhood may increase the risk of the subsequent development of a primary psychotic disorder. It is unclear whether this association is due to shared neurobiological vulnerability, common risk factors, or whether the symptoms of ADHD overtime may increase the risk of psychosis by increasing the risk of cannabis use or other substances. Five studies highlight the relationship between a previous diagnosis of ADHD and the development of psychosis later in life [8,10,11,26,27]. 

The first paper presents a narrative review on the prodromal symptoms of psychosis and then outlines a case report detailing the conditions in which ADHD symptoms may present as risk factors for psychosis [8]. In the case report, the patient demonstrated pre-psychotic characteristics of motor restlessness, impulsivity, concentration deficits, inattention, as well as non-specific prodromal symptoms [8]. Additionally, the patient reported a history of reduced concentration, attention, irritability, nervousness, loss of appetite, and unease since her childhood [8]. The patient transitioned to a pre-psychotic state as a young adult [8]. This transition was described by the patient as a process, that stemmed from self-critical thinking that later evolved to hyperreflectivity, a potential indicator of a shift from susceptibility to psychosis into a state of psychosis [8]. This patient demonstrated dichotomic thinking, paranoia, synaesthesia and catatonic symptoms eventually developed [8]. 

A systematic review of 15 studies and meta-analysis of 12 studies completed by Nourredine et al. (2021) investigated the prevalence of comorbid disorders in those with ADHD. The meta-analysis completed by the authors included 124,095 participants with ADHD and 1,725,760 controls [10]. The prevalence of psychotic disorders in cohort studies ranged from 0.4% to 6.4% while the prevalence ranged from 0 to 4.2% in the control group [10]. In the ADHD population, the prevalence of psychotic disorders ranged from 0.7% to 12.5%. The meta-analysis found that a childhood diagnosis of ADHD increased the risk of a subsequent psychotic disorder later on in life with the study reporting a relative risk of 4.74 [10].

Cassidy et al. (2011) investigated the association of cannabis use in 75 individuals with psychotic disorders, 64 with non-affective psychosis and 11 with affective psychosis [11]. The authors investigated whether childhood attention deficits are predictive of cannabis use in FEP [11]. The authors found that a childhood hyperactivity-inattention (HI) predicted inability to abstain from cannabis and lifetime cannabis-use disorder [11]. In the non-affective psychosis patients, HI predicted persistent abstinence and a diagnosis of cannabis use disorder [11]. 

Peralta et al. (2011) compared 101 patients with FEP to 21 FEP patients who met criteria for childhood ADHD [26]. This study found correlations between the severity of childhood ADHD symptoms, delay of milestones, poor academic functioning during childhood, adolescence and an earlier age of onset of psychotic symptoms [26]. This study also investigated the role of obstetric complications (OCs). Severity of childhood ADHD was related to OCs and a path analysis found that neurodevelopmental delay was a mediator between obstetric complications and ADHD symptoms [26]. 

Rho et al. (2015) compared 152 patients with FEP to 27 patients with FEP with a history of childhood ADHD [27]. The FEP patients with ADHD had significantly decreased academic achievement and had experienced the onset of psychosis two years earlier than the FEP patients without ADHD [27]. FEP patients with ADHD had poorer premorbid function than FEP patients without ADHD [27]. At 12-month follow-up, FEP patients with ADHD had significantly lower scores on the Social and Occupational Functioning Assessment Scale (SOFAS) and worse scores on the life skills profile (LSP) at baseline and 12 months [27].

### 3.5. Stimulant Use and an Increased Risk of Psychosis

Symptoms of poor attention and executive functioning deficits are shared among both ADHD and primary psychotic disorders. Executive functioning impairments are pervasive in primary psychotic disorders and tend to develop gradually, prior to the onset of positive symptoms of psychosis. For many individuals, the decline in cognitive functioning is distressing, prompting individuals to seek diagnostic assessment and treatment for their concerns. In clinical practice, it is not uncommon to have an individual present with a first episode of psychosis shortly after being started on a stimulant treatment for what is thought to be ADHD. Stimulant medications, the first line treatment for ADHD symptoms, have been shown to induce symptoms of psychosis in some individuals. At this time, it remains unknown whether or not those individuals have a decreased threshold for psychotic symptoms that is brought on by stimulant use, or whether they would have experienced a first episode of psychosis regardless of receiving treatment with stimulants or not. Regardless, this conundrum of misattributing early cognitive symptoms of a primary psychotic disorder to ADHD and then worsening the course of the illness with stimulant treatment is clinically challenging. Three papers investigated the role of stimulant use in the development of psychosis [23,24,25].

Moran et al. (2019) investigated whether the risk of psychosis in adolescents and young adults with ADHD differs between different stimulant classes [23]. A total of 337,919 people prescribed stimulants for ADHD were included in the study [23]. A total of 110,923 patients taking methylphenidate were matched with 110,923 patients taking amphetamines [23]. There was a total of 343 episodes of psychosis. In the matched population, 106 episodes (0.10%) were seen in the methylphenidate group and 237 episodes (0.21%) in the amphetamine group [23]. The median time from taking the first stimulant to psychotic episode was 128 days [23]. The author’s found that new use of amphetamine was associated with an increased risk of psychosis in comparison with new use of methylphenidate [23]. The incidence rate of psychosis was 1.78 episodes per 1000 people who were taking methylphenidate. In the amphetamine group, the incidence rate was 2.83 episodes per 1000 people [23]. 

Shibib et al., presented four cases of stimulant-induced psychosis [24]. In all four cases, patients displayed suspiciousness, distractibility, irritable and impulsive behaviour, symptoms often seen in ADHD [24]. In three of the cases, psychotic symptoms were associated with the use of a long acting formula of methylphenidate [24]. The time of onset for psychotic symptoms varied across the cases, ranging from 24 h to four months [24]. With the discontinuation of stimulant use, psychotic symptoms resolved within 24 h to 14 days [24].

A letter to the editor by Freudenreich et al. (2006) presents a case where a college student began taking and misusing stimulants to address self-diagnosed “pseudo-attention deficit disorder” [25]. The individual then became acutely psychotic after smoking phencyclidine-laced cannabis [25]. The authors hypothesize that the patient’s symptoms relating to attention, such as difficulty concentrating and declining organization skills, were due to the prodromal period of schizophrenia rather than attention deficit disorder [25]. The authors discuss the importance of recognizing symptoms that may be used to classify pseudo-attention deficit disorder as indicators of prodromal schizophrenia [25]. In this particular case, thought interference and disruptions in comprehension of language were prodromal symptoms that were attributed as symptoms of ADHD by the patient [25]. Lastly, the authors highlight stimulant use as a risk factor for some cases of schizophrenia and the highlight the importance of screening for stimulant misuse in patients presenting with self-diagnosed “pseudo-attention deficit disorder” [25].

### 3.6. Managing Comorbid ADHD and Schizophrenia

There is a lack of guidelines pertaining to the management of co-occurring ADHD and early phase psychosis. In practice, many clinicians are hesitant to prescribe stimulants in individuals with a primary psychotic disorder due to concerns of a potential worsening of positive symptoms and the risk of misuse. Two papers discussed the management of comorbid ADHD and early phase psychosis [16,19].

A narrative review by Corbeil et al. (2021) investigated the detection and treatment of comorbidities within FEP [16]. The authors found that the prevalence of ADHD in adolescents and young adults with FEP ranged from 8.1% to 24% [16]. In detecting comorbid ADHD and FEP, the authors suggest that ADHD screening should be based on a rigorous developmental history and neuropsychological evaluation [16]. For treatment, the authors suggest treatment with an antipsychotic as the psychotic disorder can contribute to and mimic symptoms of ADHD [16]. The authors flag the use of second generation antipsychotics with a lower incidence rate of extrapyramidal symptoms and psychomotor slowing [16]. Additionally the authors recommend the use of the lowest effective dose and minimizing polypharmacy, specifically anticholinergics [16]. In cases where ADHD symptoms persist, the authors suggest pharmacological treatment in combination with non-pharmacological treatments such as cognitive behavioral therapy (CBT) and cognitive remediation [16].

An article by Gough et al. (2016) discussed the management of comorbid ADHD and schizophrenia. The authors suggest to start treating the psychosis first to clarify the contribution of psychosis to the individual’s cognitive and functional impairment [19]. In cases where inattention is persistent despite treatment of psychosis, it is suggested that non-stimulant treatment options be considered such as atomoxetine or bupropion as well as non-pharmacological treatments such as CBT and psychoeducation [19]. If ADHD symptoms still persist, it is suggested to consider a stimulant trial while providing education to the patient and family on potential adverse effects [19]. In navigating concerns regarding stimulant misuse, daily medication pick-up or delivery is suggested by the authors [19]. If the use of stimulants worsens psychosis then discontinuation is recommended and although potential rechallenge could be considered in cases with severe ADHD symptoms [19].

## 4. Discussion

To our knowledge, this is the first review paper to summarize existing literature on ADHD and early psychosis. In the present paper, we reviewed the available literature on the neuropsychological profiles, rates of comorbidity, shared risk factors and clinical management of patients with comorbid ADHD and schizophrenia, focusing on early psychosis populations [7,8,9,10,11,12,13,14,15,16,17,18,19,20,21,22,23,24,25,26,27,28].

While schizophrenia is not classified as a neurodevelopmental disorder in the DSM, ADHD and schizophrenia both appear to have neurodevelopmental origins, and result in executive functioning impairments longitudinally. Several studies have investigated specific profiles of impaired cognition and compared patterns both between the two conditions, and in those with comorbid ADHD and early psychosis [7,12,13,14,15,20,21,22]. Overall, executive functioning deficits seem to be the most profound for those with early psychosis [14,15,20,21,28]. However, the severity of cognitive deficits in a specific individual would not yield useful information in terms of diagnostic clarity, although there may be certain executive functioning patterns that are more consistent with ADHD. Some studies have found that neuropsychological tests specific to measures of attentional capacity may be more sensitive in differentiating neurocognitive deficits in ADHD relative to schizophrenia [18]; however, there remains a paucity of literature on this specifically. Furthermore, in depth neuropsychological testing is not necessarily readily accessible for the majority of patients. 

ADHD can co-occur with early psychosis, and it is important for clinicians to consider a potential diagnosis of ADHD in individuals presenting with executive functioning deficits that may be in keeping with an underlying ADHD, particularly in those with a childhood history of ADHD. Beyond treatment, correctly diagnosing comorbid ADHD in an individual with early psychosis may have additional clinical implications. Individuals with ADHD and early psychosis may be at elevated risk of substance use, which may significantly impact longitudinal outcomes. There are no studies investigating whether ADHD treatment in those with comorbid early psychosis reduces the risk of substance use and associated problems. 

ADHD in childhood may itself be a risk factor for the subsequent development of a primary psychotic disorder [10,26,27]. It is unclear whether ADHD is an independent risk factor, or whether it increases the likelihood of secondary common risk factors like substance use that contribute to higher rates of psychosis. There are case reports of stimulant use being associated with the onset of psychosis in individuals who do not have a history of psychosis [24,25], and the risk appears to be higher with the use of amphetamine-based stimulants [23]. It remains unclear if there are certain individuals who are more predisposed to developing psychotic symptoms when taking a stimulant. Anecdotally, clinicians who treat psychotic disorders frequently encounter someone receiving a new diagnosis of ADHD who recently started stimulant treatment who progress overtime to meeting clinical criteria for schizophrenia. It remains unclear whether stimulant treatment in these samples precipitated the onset of psychotic symptoms, or whether a psychotic process would have evolved regardless. 

Research into ADHD and early psychosis is lacking in many respects. There are no scales or objective tests for the diagnosis of ADHD that have been validated in early psychosis samples. The ability to apply a validated test, or even specific criteria to facilitate making an accurate diagnosis of ADHD in individuals with a history of psychosis would be beneficial to both patients and clinicians. Tools to aid diagnostic certainty are of particular relevance given the potential association of stimulant use and the onset of psychotic symptoms. Furthermore, there are no studies or even general clinical guidelines regarding clinical management of ADHD in someone with early psychosis. Studies evaluating the use of stimulant treatment for those with both conditions to assess both risks and benefits are warranted. In addition, research should investigate the clinical utility of non-medication strategies given the issue of potential worsening of the underlying psychotic disorder. While many clinicians are concerned about the theoretical risk of worsening positive psychotic symptoms with the addition of stimulant treatment, it is unclear how often this occurs in those who continue to take antipsychotic maintenance treatment. 

In terms of limitations of this work, there is overall a lack of research addressing ADHD and early psychosis. Therefore, many of the papers discussed pertain to variable topics within ADHD employing differing research methods, making it challenging to generate overall conclusions. Furthermore, we did not use a systematic review to explore the relationship between ADHD and early psychosis. Our intention was to gain a broad appreciation of the existing research with this narrative review however, a systematic review or meta-analysis would facilitate answering more specific questions related to ADHD and early psychosis. Lastly, the classification of ADHD varied across the studies. While some studies utilized formal diagnostic criteria to define ADHD, one study identified ADHD using a self-report scale [9]. For future studies, it will be important that the identification of ADHD is consistent in order to accurately compare and contrast results amongst these studies. 

## 5. Conclusions

Overall, research investigating ADHD and early psychosis is sparse and much remains unknown. Future studies should delineate how to differentiate ADHD from executive functioning secondary to early psychosis, investigate whether or not stimulant medications improve ADHD symptoms in those with early psychosis and importantly, explore the risk of negative outcomes, including worsening of the primary psychotic illness. Stimulant treatment likely confers an elevated risk of psychosis, but it remains unclear whether certain subgroups are more at risk, and should not be offered stimulant treatment, or monitored more closely. Understanding the nuances related to managing co-occurring ADHD and early psychosis would lead to improved clinical care for patients, and the ability for clinicians to make treatment recommendations based on evidence. 

## Figures and Tables

**Figure 1 brainsci-14-00190-f001:**
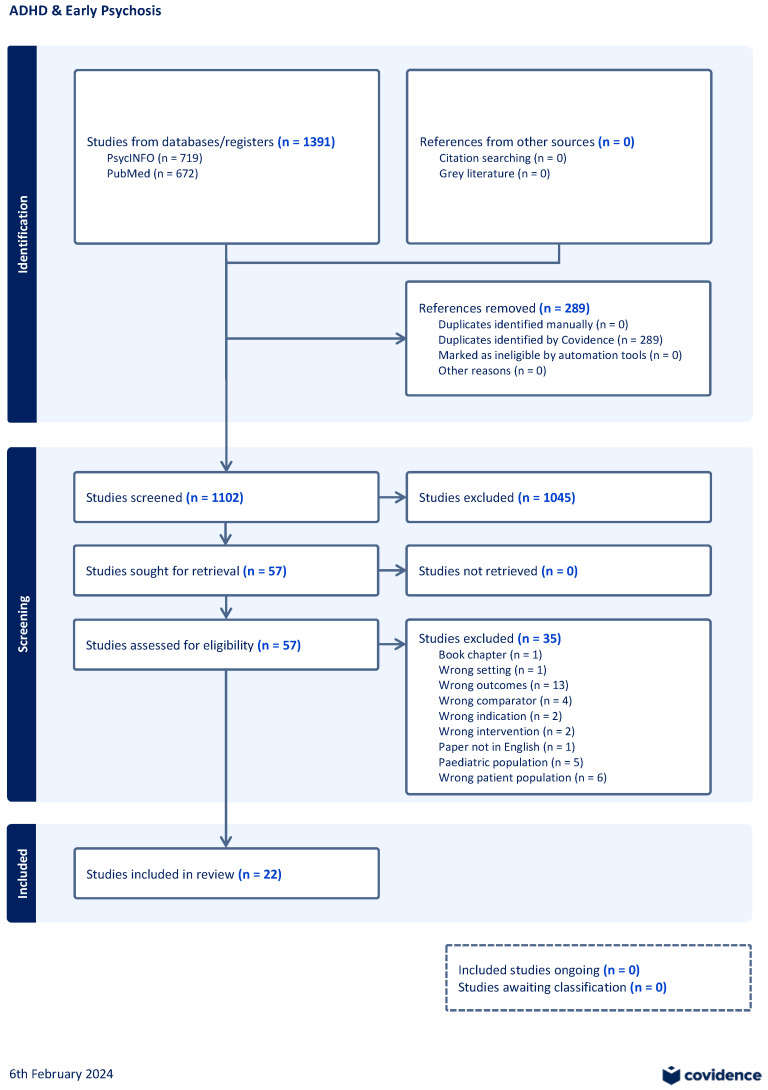
Schematic of screening process.

## Data Availability

Not applicable.

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
