# Peer review of "A Narrative Review Exploring Attention Deficit/Hyperactivity Disorder in Patients with Early Psychosis"

_brainsci, 2024, doi:10.3390/brainsci14030190_

Round 1
Reviewer 1 Report
Comments and Suggestions for Authors
This article by Toba-Oluboka and Dempster aims to explore and summarize the existing literature on ADHD and early psychosis. The topic is undoubtedly relevant and timely considering that, despite the high prevalence of the disorder in the younger population, the link between ADHD and early psychosis is still relatively unexplored.
As a whole, the paper is well presented, with a fluid text and appropriate vocabulary, although some minor mistakes (typos, missing commas) are present. Each section of the manuscript is pertinent and in line with the purpose of the Authors, although the background should be enriched with a more detailed explanation of the relationship between ADHD and early psychosis.
While acknowledging the great work done by the Authors and the unquestionable quality of this manuscript, there are some major and minor observations that they might be willing to address.
Major issues
· Abstract and Introduction. There are few statements that should be supported by a reference.
o “schizophrenia is a neurodevelopmental disorder”: this is not in line with the current classification in major diagnostic systems (DSM, ICD); indeed, the neurodevelopmental hypothesis of schizophrenia should be presented as a theory that has been implemented with the most recent advances in research on this topic. For this purpose, see this paper by Murray and colleagues (Murray RM, et al. 30 Years on: How the Neurodevelopmental Hypothesis of Schizophrenia Morphed Into the Developmental Risk Factor Model of Psychosis. Schizophrenia Bullettin. 2017).
o “stimulant use has been associated with the onset of psychotic symptoms in some individuals, but it is unclear whether certain subgroups are more susceptible”: is there a reference for this statement? i.e. Curran C, Byrappa N, McBride A. Stimulant psychosis: systematic review. British Journal of Psychiatry. 2004.
· Introduction. I suggest enriching the background of this review in order to substantiate the importance of this article to the reader, as recommended by the SANRA guidelines (Scale for the Assessment of Narrative Review Articles; Baethge et al., 2019).
· Introduction. A clearly stated and defined research question should be made explicit in this section. For this purpose, I recommend keeping the PICO model in mind, in order to clearly identify the outcome(s) of this literature synthesis.
· Materials and Methods. PRISMA statement is not the reference that is best suited of this type of paper. Although narrative reviews do not require a rigorous methodology, I advise the Authors to refer to the recently approved SANRA guidelines (Scale for the Assessment of Narrative Review Articles; Baethge et al., 2019).
· Figure 1, flowchart. First, the flowchart should be moved to the appropriate section. Furthermore, numbers of excluded studies should be indicated in the appropriate part (i.e., n=).
Minor comments
· In the flowchart (figure 1) the Authors cite an initial selection of 1391 studies, later to become 1102; however, this passage is not made explicit in the Materials and Methods: I suggest adding it for clarity.
· In the Materials and Methods section the Authors state that, of the 57 studies initially selected, 35 were excluded; the account, however, does not add up, as the studies excluded due to ‘wrong indication’( n=2) have been left out.
Comments on the Quality of English Language
· On Line 198 and 333 there are typos: please correct “comonly” with commonly and “cormobid” with comorbid.
Author Response
Dear Reviewer,
Thank you for your comments and suggestions on our manuscript. Below outlines how we have addressed your comments:
“schizophrenia is a neurodevelopmental disorder”: this is not in line with the current classification in major diagnostic systems (DSM, ICD); indeed, the neurodevelopmental hypothesis of schizophrenia should be presented as a theory that has been implemented with the most recent advances in research on this topic. For this purpose, see this paper by Murray and colleagues (Murray RM, et al. 30 Years on: How the Neurodevelopmental Hypothesis of Schizophrenia Morphed Into the Developmental Risk Factor Model of Psychosis. Schizophrenia Bullettin. 2017).
- We agree with this comment and have reworded this sentence in the introduction (lines 26-27) and have added in the suggested reference.
“stimulant use has been associated with the onset of psychotic symptoms in some individuals, but it is unclear whether certain subgroups are more susceptible”: is there a reference for this statement? i.e. Curran C, Byrappa N, McBride A. Stimulant psychosis: systematic review. British Journal of Psychiatry. 2004.
- We agree that this idea needed to be cited by the literature. As the sentence mentioned in this comment is in the abstract, we added the recommended reference into our introduction(line 49). We did not add the citation to the sentence in the abstract as the abstract does not usually contain citations.
Introduction. I suggest enriching the background of this review in order to substantiate the importance of this article to the reader, as recommended by the SANRA guidelines (Scale for the Assessment of Narrative Review Articles; Baethge et al., 2019).
- In addressing this comment, we reviewed the SANRA guidelines. We decided to add more information on symptoms of ADHD and how these symptoms relate to psychosis. We believe that this will strengthen the importance of the article to the reader.
Introduction. A clearly stated and defined research question should be made explicit in this section. For this purpose, I recommend keeping the PICO model in mind, in order to clearly identify the outcome(s) of this literature synthesis.
- We agree with the reviewer that a clearly stated research question is important. After consulting the SANRA guidelines where it is suggested that a clearly stated aim or question is important to include for this type of review, we decided to revise lines 53-54 to include a clear aim of our review. We chose to state an aim rather than a question because our review is on a topic that is underreported in the literature, making it hard for us to ask a specific question about this topic.
Materials and Methods. PRISMA statement is not the reference that is best suited of this type of paper. Although narrative reviews do not require a rigorous methodology, I advise the Authors to refer to the recently approved SANRA guidelines (Scale for the Assessment of Narrative Review Articles; Baethge et al., 2019).
- We agree that the SANRA guidelines are more appropriate for this type of review. We have amended this sentence in the methods section (line 56).
Figure 1, flowchart. First, the flowchart should be moved to the appropriate section. Furthermore, numbers of excluded studies should be indicated in the appropriate part (i.e., n=).
- We have moved the flowchart to the methods sections where it is referenced in the text and have included the numbers of excluded studies in n= format in the figure and text (lines 75-77).
In the flowchart (figure 1) the Authors cite an initial selection of 1391 studies, later to become 1102; however, this passage is not made explicit in the Materials and Methods: I suggest adding it for clarity.
- We agree with the reviewer and have added this into our materials and methods section (Lines 71-72)
In the Materials and Methods section the Authors state that, of the 57 studies initially selected, 35 were excluded; the account, however, does not add up, as the studies excluded due to ‘wrong indication’( n=2) have been left out.
- We have added "wrong indication (n=2).
Thank you again for you comments on our manuscript. We appreciate your time and effort in improving this paper.
Kind Regards,
Ms. Toba-Oluboka & Dr. Kara Dempster
Reviewer 2 Report
Comments and Suggestions for Authors
This paper by Toba-Oluboka and Dempster deals with two important diagnoses in child psychiatry: ADHD and psychosis. Through this narrative review the authors try to describe how these two conditions are entangled, highlighted common substrates and differences. The manuscript is well written, enjoyable to read, scientifically valid, and original in its premises. I have no requests for the authors, except for a little suggestion: I would put the headings of the sub-paragraphs in the results section as affermative sentences, rather than questions.
Author Response
Dear Reviewer,
Thank you for your kind words and feedback. We have changed our subheadings in the results section to affirmative sentences.
We appreciate your time and effort in improving our manuscript.
Ms. Toba-Oluboka & Dr. Dempster
Reviewer 3 Report
Comments and Suggestions for Authors
The topic of the manuscript is very important, and it contains relevant information about ADHD and early psychosis. According to Authors this is the first review paper to summarize existing literature on ADHD and early psychosis. The review (except for the section of the Introduction) is well written and comprehensive.
There are minor points which need to revise.
-The section of the Introduction is too short, not informative and it does not demonstrate the possible relationship between development of symptoms of the ADHD and psychosis. The materials and methods, especially the section of results is well described with enough and wide information. The results section contains very good questions and hypothesis, as well as many relevant literature references. The section of Discussion is excellent, includes many greatly well interpreted information. The Conclusions states the problem and contains relevant recommendations.
In my opinion, it is very well written review and I support its publication after minor correction.

Author Response
Dear Reviewer,
Thank you for your kind words and feedback on our manuscript. To address your comments on our introduction, we have added more information in our introduction on ADHD symptoms and how these symptoms relate to psychosis.
Thank you for your time and effort in improving our manuscript.
Ms. Toba-Oluboka & Dr. Dempster
Reviewer 4 Report
Comments and Suggestions for Authors
This study reviewed the papers examining the associations between attention-deficit/hyperactivity disorder (ADHD) and early psychosis, especially the roles of stimulants for the occurrence of psychotic symptoms.
I have some suggestions for the authors to improve this manuscript.
1. At the beginning of the article, it mentions schizophrenia; but early psychosis and psychotic disorders were also used in the text. The three terms “schizophrenia,” “early psychosis,” and “psychotic disorders” have different definitions. The authors need to explain why they mixed use of them.
2. Although all “schizophrenia,” “early psychosis,” and “psychotic disorders” originate from neurodevelopmental problems, they are not contained in neurodevelopmental disorders in DSM-5-TR. Therefore, the sentences such as “ADHD and schizophrenia are neurodevelopmental disorders…” warranted revisions.
3. Citations should be added properly. For example, “the use of stimulants, which are the first line treatment for ADHD [4], has been hypothesized to increase the risk of positive psychotic symptoms, like hallucinations and delusions.”
4. Full spellings should be added when the abbreviations appeared for the first time. For example, “ADHD” in Abstract.
5. The title should be revised into “A Narrative Review Exploring Attention-Deficit/Hyperactivity Disorder in Patients with Early Psychosis.” The same, the descriptions in Abstract and text should be revised.
6. Abstract: ”This risk may be modified by higher rates of substance use or could be related to shared premorbid risk factors for both conditions, such as obstetrical complications.” Should “or” be replaced by “and”?
7. Introduction: “...first-line dopamine blocking treatments…” Pure dopamine receptor blockers are not the first-line drugs for psychotic disorders such as schizophrenia.
Comments on the Quality of English LanguageGood
Author Response
Dear Reviewer,
Thank you for your feedback on our manuscript. Below we outline how we have addressed your comments.
At the beginning of the article, it mentions schizophrenia; but early psychosis and psychotic disorders were also used in the text. The three terms “schizophrenia,” “early psychosis,” and “psychotic disorders” have different definitions. The authors need to explain why they mixed use of them.
- We agree with the reviewer that the use of “schizophrenia,” “early psychosis,” and “psychotic disorders” is confusing as they do all have different definitions. We have gone through the manuscript and have used "early psychosis" to make the terminology consistent throughout our paper.
Although all “schizophrenia,” “early psychosis,” and “psychotic disorders” originate from neurodevelopmental problems, they are not contained in neurodevelopmental disorders in DSM-5-TR. Therefore, the sentences such as “ADHD and schizophrenia are neurodevelopmental disorders…” warranted revisions.
- We have revised the sentence in lines 383-385 to address this comment.
Citations should be added properly. For example, “the use of stimulants, which are the first line treatment for ADHD [4], has been hypothesized to increase the risk of positive psychotic symptoms, like hallucinations and delusions.”
- We have moved the citation to the end of the sentence (lines 45-46).
Full spellings should be added when the abbreviations appeared for the first time. For example, “ADHD” in Abstract.
- We have made this change in the abstract (line 7).
The title should be revised into “A Narrative Review Exploring Attention-Deficit/Hyperactivity Disorder in Patients with Early Psychosis.” The same, the descriptions in Abstract and text should be revised.
- We have changed the title of the manuscript to the suggested title. And we have revised the descriptions in the abstract and introduction.
Abstract: ”This risk may be modified by higher rates of substance use or could be related to shared premorbid risk factors for both conditions, such as obstetrical complications.” Should “or” be replaced by “and”?
- We agree with the reviewer and have replaced "or" with "and".
Introduction: “...first-line dopamine blocking treatments…” Pure dopamine receptor blockers are not the first-line drugs for psychotic disorders such as schizophrenia.
- We have reworded this to clarify our sentence in the introduction (lines 34-35).
Thank you again for your comments on our manuscript. We appreciate your time and effort in improving our manuscript.
Ms. Toba-Oluboka & Dr. Dempster